# Protocol for black student well-being study: a multi-site qualitative study on the mental health and well-being experiences of black UK university students

Nkasi Stoll [1,2] Yannick Yalipende,[3] Jason Arday,[4] Dominic Smithies,[5] Nicola C. Byrom,[6] Heidi Lempp,[7] Stephani L. Hatch[1,2]

**Correspondence to**
Nkasi Stoll;
nkasi.1.stoll@kcl.ac.uk

## ABSTRACT

**Introduction** There is an increasing concern about the mental health and mental well-being of university students in the UK. Black university students who report a mental health condition are less likely to complete their course, achieve a first-class or upper second-class degree and progress to further education. This study will document black university students' accounts of their mental health experiences and perceptions of key turning points of biographical changes to their mental health as they move through the university life cycle.

**Methods and analysis** This is a qualitative study. Data will be collected through a biographical narrative interpretive method. Interviews will enable the researcher to study systematically how participants make sense of themselves and account for the complexities of their life experiences, from their own perspectives and language. An interpretative phenomenological approach will be used to offer insights into what black students studying at UK universities report affects their mental health and well-being. Data collection for this study commenced in October 2020. Data collection and analyses will be completed by January 2022.

**Ethics and dissemination** Full ethical approval for the current study was obtained from King's College London Psychiatry, Nursing and Midwifery Research Ethics Subcommittee (Rec Ref: 20489, Project Ref: HR-19/20-20489, 2 October 2020). From the study findings, we aim to contribute to the evidence base, make recommendations for interventions and encourage further study into black student mental health.

## Strengths and limitations of this study

► The current study is designed to be the first UK national multi-site study to explore, from black undergraduate and postgraduate university students' perspectives, the life events and experiences that affect their mental health and well-being throughout the university life cycle.

► A multidisciplinary coding team (including black university students) will ensure the inclusion of multiple perspectives to reduce researcher bias and provide opportunities to discuss disagreements in interpretations.

► This study will contribute to the evidence base on race, mental health and higher education; provide relevant recommendations for interventions; and encourage further study into black student mental health.

► The current project is exploring the lives of 20–25 black students studying at 10 UK universities, so may lack generalisability, representativeness and transferability of findings.

time that many enter and navigate university in the UK. There are approximately 2.4 million university students in the UK, the majority being young adults,[10] which means Higher Education Institutes (HEIs) are in the best position to provide prevention and intervention to many young adults during an important transitional period.

Throughout the university life cycle,[11] students are exposed to a range of experiences that make higher education a high-risk period for maladaptive coping and possible onset of mental health problems, including (but not limited to): individuation, separation from family, increased autonomy and responsibility, academic-related stress, financial concerns, sleep disruption, balancing conflicting demands of studying

## INTRODUCTION

The mental health and mental well-being of university students in the UK is of increasing concern to policymakers, researchers, charities, university staff and students.[1–4] Young adulthood is a critical age for emotional, psychosocial and cognitive development[5][6] and first onset of mental health problems, suicidality and substance misuse problems.[7–9] Young adulthood is also the

with personal and family life, and exposure to risky behaviours, including recreational drug use and alcohol binging.[12–16] Unsupported mental health problems are associated with progression to other comorbid disorders, substance use disorders, self-harm and suicide ideation and attempts.[17–20]

A growing body of evidence explores inequalities and inequity in student admissions, experiences and outcomes in UK HEIs by race, gender, class and (dis)ability.[21–27] Black students with a mental health problem studying at UK HEI are less likely to complete their course, achieve a first-class or upper second-class degree and progress to further education.[28 29] The Office for Students[2] reported that 'black students with mental health conditions are being failed throughout the student cycle' (pg. 6), but that there are limited data or research available to explain how and why. Available research indicates the following factors and dimensions may explain the inequality and inequity observed in higher education, at the expense of black students: racism and structural racism, discrimination, mental health stigma, sexism, cultural incompetence and insensitivity of healthcare professionals.[30–32] It is essential to know more about the experiences of, risk factors for, and challenges that affect black university students' mental health in UK higher education in order to (re-)design, develop and deliver appropriate mental health and well-being services.

### Critical race theory

Critical race theory (CRT) offers an interpretive theoretical approach to explore and challenge racial inequality in society. It is based on the understanding that racism is embedded as normal practice within society and institutions, rooted in slavery and colonialism; and race is a socially constructed concept that is used by white people to further their socioeconomic and political interests and power at the expense of racialised minorities. This racial bias causes discrimination within law, employment, housing, healthcare, politics and education, disadvantaging racialised minorities.[33 34] CRT in the educational context explores and challenges the white supremacist patriarchal structures and assumptions that have historically shaped education.[33 35] CRT theorists explore racial inequality in: admissions, curriculum and pedagogy, teaching and learning, institutional culture, campus racial climate, and policy and finance within UK further and higher education.[36–40] CRT is also used as a conceptual lens through which to interrogate the generational impact slavery and colonialism has on an individuals and families, as well as psychological research and practices, at the expense of racialised minorities.[41 42] Therefore, CRT will guide the direction of this current study by providing a framework to support the study team to interrogate the methodology, analyses and interpretation throughout the research process.[32]

### Aims

The aim of this study is to document and explore black university students' (1) accounts of their mental health experiences as they transition throughout their university lives; and (2) perceptions of key turning points of biographical change to their mental health in their lives as they move through the university student life cycle.

### Objectives

This study uses biographic narrative interpretive method (BNIM)[43 44] interviews and interpretative phenomenological approach (IPA)[45–52] to get an in-depth understanding of the educational and mental health of experiences black UK university students.

### Research questions

1. From the students' perspective, what life events and experiences do black university students perceive affect their mental health and well-being?
2. How does institutional and structural racism within UK education systems affect the mental health and well-being experiences and outcomes of black university students?

## METHODS AND ANALYSIS
### Study design

This is a qualitative study. Data are being collected through BNIM interviews.[43 44] This interview technique enables the researcher to study how participants make sense of themselves and account for the complexities of their life experiences, from their own perspectives and choice of language.[43 44] For the current study, IPA[45–52] offers insights into what black students studying at UK universities report affects their mental health and well-being. Data collection for this study commenced in October 2020. Data collection and analyses are planned to be completed by January 2022.

### Recruitment

A webpage was set up specifically for the study on the Health Inequalities Research Network (HERON) website: www.heronnetwork.com which is General Data Protection Regulation (GDPR) complaint. The webpage has the lead researcher (NS)'s contact information, the study's rationale, expression of interest form, information sheet, consent form, list of support services and updates about the timeline of the project. The study is advertised on social media accounts (Twitter and Instagram) set up specifically for the study. Social media posts with a short summary of the study were designed and sent out on this social media page to recruit participants. The social media posts include the study's website and QR code which means that potential participants can access the webpage with one click. Recruitment emails were sent to student unions, student groups and student services.

Completed expression of interest forms are automatically sent to a King's College London (KCL) study mailbox which was set up for the study. NS monitors the incoming forms two times a day and screens the participants based on the inclusion and exclusion criteria. Potential participants are contacted by NS via email

to complete an attached consent form which they are asked to email back to the researcher. Participants have a minimum of 48 hours to decide whether they would like to participate in the study. If the participant has not responded after 2 days they are sent a reminder email by NS and subsequent two reminder emails until they are considered non-responsive.

## Sample size

Purposive sampling[51] is being used to recruit black university students who meet the inclusion criteria. Fifteen to twenty students are being recruited from nine Russell and non-Russell group universities across the UK. Data from the Higher Education Statistics Agency[10] for UK higher education student enrolments by higher education provider and ethnicity for the academic years 2014–2015 to 2018–2019 were used to decide which universities to recruit from. In Microsoft Excel, the data were sorted into higher education providers with the highest percentage of black students enrolled, to the lowest. After this, IQR was calculated to divide the higher education providers into quartiles of median, lower (ie, lowest percentage of black students) and upper range (ie, highest percentage of black students). Three higher education providers that fall into each range were chosen.

## Sample

Two universities are in Ireland, one university is in Scotland and seven universities are in England. Of the universities based in England, three are in London, one is in the East Midlands, one university in Southeast England, one in the West Midlands and the final in East of England. Five universities are Russell Group universities and five are non-Russell Group universities. Russell group universities are defined as 'a group of units with a shared focus on research and a reputation for academic achievement'.[53] While data from the National Student Surveys suggest there are no statistically significant differences in student satisfaction between Russell and non-Russell group universities.[54] There is some evidence that ethnic minority applicants to Russell Group universities may be less likely to receive offers to study, compared with similarly qualified white applicants.[55] Six universities are based in the city where the campuses and facilities are spread out across the city, and four universities are campus universities, situated in rural settings.

## Inclusion and exclusion criteria

The inclusion criteria for students are deliberately broad to include students who: (1) self-identify as black (Caribbean, African and mixed heritage with black); (2) are currently studying at, within a year of graduating from, or are within a year of dropping out of a university course; (3) self-identify as having struggled with their mental health and/or well-being while studying at a UK university; (4) are aged 18 or over. As an IPA study,[45–52] this will allow for a wide range of black university students, from different backgrounds, to be eligible for study inclusion as the study processes.

The exclusion criteria are people who: (1) have never been enrolled on a course at a UK university, (2) are enrolled at a university outside of the UK, (3) do not identify as black (Caribbean, African and mixed heritage with black), (4) graduated or dropped out of university over a year ago, (5) are aged 17 or younger.

Potential participants are asked to complete an expression of interest form online, with sociodemographic information to determine their eligibility for the study.

## Data collection

Data collection for this study commenced in October 2020. Data collection and analyses will be completed by January 2022. All interviews are being conducted by NS. In order to generate a narration that allows the interviewee to begin, construct and end their narrative on their own terms, students are invited to participate in a hybrid adapted version of (BNIM)[43 44] designed by NS in collaboration with the authors'. The BNIM approach to data gathering is an open narrative interview process where the interviewer is encouraged to participate in active listening, and not interrupt with questions. The students are asked a single question called a 'Single Question Inducing Narrative' (SQUIN)[43 44] to describe their life, in terms of their mental health and well-being. The SQUIN reads as follows: 'As you know, I am researching mental health and well-being experiences of black university students in the UK. I understand that you have had such experiences. So please can you tell me the story of your mental health experiences before university and while studying at university. Please include all the events and experiences that were important for you, personally, up until now. There's no rush, you can start wherever you like. We've got 2 hours in total. I'll listen first, I won't interrupt. I'll tell you if we are running out of time. I'll just take some notes and when you've finished telling me about it all we will have a 15 min break. After the break I may ask you further questions about your story'. As the student is talking, NS takes notes on what Wengraf terms 'Particular Incident Narratives' (PINs),[43 44] which are significant events that have occurred in each student's life surrounding their mental health and well-being using the keywords that they use. Students are then invited to take a 15-min break and during this time NS identifies questions called narrative/N-pointed questions,[43 44] to clarify or pick up on PINs.[43 44] The second session begins with NS asking the student to reflect back on the identified PINs and continue with the N-pointed questions. The interview ends with the question: 'Before we finish, is there anything you would like to add that we didn't talk about?'. Interviews last up to 2 hours in total. The interviews elicit two transcripts of narrative for data analysis from each student.

Interviews are conducted virtually using Microsoft Teams and are recorded with the student's permission. NS transcribes the interviews by hand using Microsoft

Teams' automated speech-to-text transcript programme, before analysis.

A BNIM pilot study was conducted to assess comprehension of the BNIM question, relevance, timing and research burden.[56] Two black University students were invited to participate in the pilot study. Data were presented to the fifth, sixth and seventh authors of this paper to collaboratively agree whether they matched the study objectives. No amendments were made to the study.

### Data analysis

Data analysis began after the first interview in October 2020. Data are being analysed using IPA[45–52] on NVivo Qualitative Data Analysis Software (V.12). An inductive analysis of each student's account is conducted before moving onto group analysis.[47] Analysis began with multiple readings of each student's transcript by NS, who makes notes on the student's language and structure (including repetitions, exaggerations, emphasis, symbols, pauses, slang), emotional responses and any insights into the students' experience and perspective on their world, that is, 'codes'. This is to interpret what and how the student is communicating their story. As the analysis develops, NS looks for patterns in the codes to create emerging themes. Themes identify recurring patterns of meaning (ideas, thoughts, feelings, topics of importance or concern) presented by the student and interpreted by NS throughout the text. Themes will eventually be linked and grouped together to create subordinate themes. The final set of themes will be summarised, arranged into clusters to form master themes and placed into a table. Quotations from the students will be selected to illustrate each theme. All the transcripts will be collated and re-examined by NS who will pay attention to thematic overlap and differences in student's accounts and will make links to existing psychological and sociological literature, concepts and theories which provide an in-depth understanding of the phenomena and strengthen the robustness of the data analysis. The authors will be careful to apply a critical race lens on the existing theoretical knowledge within this area of research to explain the phenomena.[55]

In IPA, 'the participants are trying to make sense of their world; the researcher is trying to make sense of the participants trying to make sense of their world' (pg. 53).[51] The dialogue between the researcher and the participant is described as a double hermeneutic and IPA recognises the importance of constant self-awareness and reflexivity.[57] NS writes notes in a reflective diary throughout study conception to dissemination to draw their observations and reflections about the interview experience, data analysis and interpretations; or any other thoughts and comments of potential significance.[57–59]

A second coder, a black university student (YY), will code 20% of the interviews using IPA for cross-validation of the themes.[60 61] The authors of this paper will form a multidisciplinary coding team and participate in conversation circles over Microsoft Teams to validate the data, participate in theme development and promote reflexivity and dialogue.[60 62] The authors all have extensive knowledge and experience on student mental health and mental health inequality in the UK. The circle mediator (NS) will participate in the conversation to present and clarify ideas or issues and encourage the dialogue. The conversation circle will be audio recorded with verbal consent of the attendees. During the conversation, the objectives, methodological process and research results will be presented. The authors will be encouraged, in a dialogic process, to give their opinion about the codes and emergent themes. A second conversation circle will be arranged for the subordinate themes.

### Data protection

All confidential data are kept for 10 years according to KCL Records and Data Retention Schedule (2019).[63] Data are stored on password-protected computers in password protected, encrypted folders on KCL Share-Point which are only identifiable by the code number and only accessible to members of the research team. During transcription, audio-recordings are anonymised, with all identifiable information removed prior to using NVivo software. All audio-recordings are destroyed immediately after transcription. This is important to ensure that their identities are protected, and they are not at risk of harm due to damage of their reputation or social standing, for example, should they express controversial or negative opinions or experiences. This is particularly relevant considering the sensitive topic of study and the fact that participants are invited to consider current as well as historical experiences at university.

### Public and patient involvement

The lead author is a black Caribbean PhD student who designed and wrote up the study protocol and the second author is a black African master's student who was involved in the design of this protocol.

### ETHICS AND DISSEMINATION

Full ethical approval has been granted by KCL Psychiatry, Nursing and Midwifery Research Ethics Subcommittee (Rec Ref: 20489, Project Ref: HR-19/20-20489, 2 October 2020). Written informed consent to participate and audio-recorded interviews is obtained from all participants. Participants consent to the inclusion of material pertaining to themselves, that they acknowledge they cannot be identified via the manuscript; and that the participant has been fully anonymised by the author. Data management and storage is subject to the UK Data Protection Act 1998 and GDPR. The authors assert that all procedures contributing to this work comply with the ethical standards of the relevant national and institutional committees on human experimentation and with the Helsinki Declaration of 1975, as revised in 2008.

### Safety considerations

Potential risks of participating come as a result of the participant disclosing a historical or current distressing

experience. During the consent process and at the beginning of the interview, the interviewer (NS) informs each participant that confidentiality needs to be broken if they disclose personal details which raises concerns to NS about their safety, the safety of others, safety of someone under the age of 18 or about a previously non-disclosed serious crime they or someone else has committed or is about to commit.

When a participant discloses a serious immediate risk, the interview is stopped, a conversation is had with the participant (where appropriate) to ensure they understand why and how information is shared for their safety, and the emergency services is contacted immediately. The co-senior author (SH) is informed immediately after (in person, or by phone if this is not possible). Incidents are reported to the chair of the ethics committee.

When a participant becomes distressed during the interview NS asks the participant if they would like a break. When the participant would like to terminate the interview or NS decides the participant is no longer able to continue she offers to re-schedule the interview at the next convenient time.

Time is explicitly allocated to a 'debrief' period at the end of the interview for all participants to reflect on the session, ask any questions or raise anything that may be of concern. Participants are asked: 'Is there anything you would like to share regarding your experience talking to me today?'. A list of support services available across the UK is listed in a document created for this research project, which is available on the research website in case participants need professional support. Every participant is sent an email directing them to this list of support services after the interview.

### Output and dissemination

Findings will be disseminated via an internal report (doctoral thesis), peer-reviewed journal articles (eg, commentary piece(s), theoretical article(s), write up of the main findings) and conference presentations (eg, Advance HE, The Student Mental Health Research Network, AMOSSHE, The Office of Students). The study team will seek funding from the London Interdisciplinary Social Science Doctoral Training Partnership, KCL and the ESRC Centre for Society and Mental Health to design and deliver internal and external public engagement events to discuss the study findings and potential recommendations for interventions. An anthology book will be written as a collection of the participant's stories, the lead researcher's story and the lead researcher's reflections of the stories. The purpose of this anthology is to disseminate the research findings to people interested in black university student mental health, including current students and academic staff, families of students, mental health practitioners, community organisations, policy-makers, advocates and charities. From the study findings, we aim to contribute to the evidence base on race, mental health and higher education; provide relevant recommendations for interventions and encourage further study into black student mental health.

### DISCUSSION

To our knowledge, the current study is designed to be the first multi-site UK study to explore, from black undergraduate and postgraduate university students' perspectives, the life events and experiences that affect their mental health and well-being throughout the university life cycle. This qualitative approach has not been previously employed to understand what and how individual, institutional and structural factors within UK education systems impact the mental health and well-being experiences of black university students. The usefulness of the data and subsequent recommendations are dependent on what is derived from the data. A multidisciplinary coding team will ensure the inclusion of multiple perspectives in order to reduce researcher bias and provide opportunities to discuss disagreements in interpretations.[60] IPA is one of the appropriate methodological choices to offer insights into how a person, in a particular context, makes sense of a given phenomena.[64 65] Combining narrative inquiry with phenomenology and hermeneutics is a suitable approach to attempt to gain insight into black students' subjective experiences of life in university, while at the same time encouraging black students to tell their own stories and have their voices heard.

The study findings will be limited to the time and context of this study as realities and perspectives will differ across contexts and time.[66] Interpretation of study findings will take into account potential limitations regarding the generalisability of the findings. The current project is exploring the lives of 20–25 black students studying at 10 UK universities, so the findings are not generalisable, representative and transferable to all black university students studying at UK HEIs . However, in IPA research, 'samples in IPA studies are usually small, which enables a detailed and very time-consuming case-by-case analysis' (pg. 4).[57] IPA is not concerned with generating a generalisable theory, however, comparisons of multiple IPA studies on the mental health of black university students and student mental health inequalities over time may provide insights into generalisable patterns and mechanisms.[51 57] Future research may include a mixed-methods study of black students across the UK educational pipeline, as well as further exploration of the mental health experiences of black students who face multiple marginalisation, for example, disabled; first-generation; carers and lesbian, gay, bisexual, transgender and other queer-identifying community (LGBT+) black students.

**Author affiliations**
[1]Department of Psychological Medicine, Institute of Psychiatry, Psychology and Neuroscience, London, UK
[2]Centre For Society and Mental Health, King's College London, London, UK
[3]Division of Psychology and Language Sciences, University College London, London, UK
[4]Department of Sociology and Social Policy, University of Durham, Durham, UK

[5]Student Minds, Leeds, UK
[6]Department of Psychology, King's College London, London, UK
[7]Inflammation Biology, King's College London, London, UK

**Acknowledgements** The authors would like to thank Dr Tom Wengraf for his contribution by reading and commenting on the initial draft of the manuscript.

**Contributors** NS conceived, designed and wrote the protocol. SH, HL and NCB were involved in the design, supervision and contributed to the final manuscript. YY, DS and JA contributed to the final manuscript.

**Funding** NS is supported by Economic and Social Research Council (ESRC) (grant number ES/P000703/1) via the London Interdisciplinary Social Science Doctoral Training Partnership (LISS-DTP) and the ESRC Centre for Society and Mental Health at KCL (ES/S012567/1). SH is partly supported by the ESRC Centre for Society and Mental Health at KCL (ES/S012567/1) and by the the National Institute for Health Research (NIHR) Maudsley Biomedical Research Centre at South London and Maudsley NHS Foundation Trust and King's College London (BRC-1215–20018). SH also currently received funding from the Wellcome Trust (203380/Z/16/Z) and the Economic and Social Research Council (ESRC) (ES/V009931/1). The views expressed are those of the authors and not necessarily those of the NHS, the NIHR or the Department of Health and Social Care, Wellcome Trust, ESRC or KCL. HL currently receives funding for successful grants as a PI or co- PI: Wellcome Trust Institutional Strategic Support Fund (No grant number); King's Health Partner Multiple Long-Term Conditions Challenge Fund (No grant number); National Axial Spondyloarthritis Society (No grant number); UKRCI Medical Research Council (MR/S001255/1) and (MR/R023697/1); NIHR (RP-PG-0610-10066); Guy's and St. Thomas' Charity, London (EFT151101); and vs Arthritis (No grant number). NCB is partly supported by the ESRC funding for SMaRteN (ES/S00324X/1).

**Competing interests** None declared.

**Patient and public involvement** Patients and/or the public were involved in the design, or conduct, or reporting, or dissemination plans of this research. Refer to the Methods section for further details.

**Patient consent for publication** Not applicable.

**Provenance and peer review** Not commissioned; externally peer reviewed.

**ORCID iD**
Nkasi Stoll http://orcid.org/0000-0003-0427-3367

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
