## [Reviewer comments · BMJ Open]

ARTICLE DETAILS

TITLE (PROVISIONAL)	Protocol for Black Student Wellbeing Study: A multi-site qualitative study on the mental health and wellbeing experiences of Black UK university students
AUTHORS	Stoll, Nkasi; Yalipende, Yannick; Arday, Jason; Smithies, Dominic; Byrom, Nicola; Lempp, Heidi; HATCH, STEPHANI

VERSION 1 – REVIEW

REVIEWER	Leath, Seanna University of Virginia
REVIEW RETURNED	28-Sep-2021

GENERAL COMMENTS	 • I'd reconsider some of the examples that authors offer as contributing to mental health problems – mainly just the “new social connections” part – the others make sense without further explanation • CRT does seem useful for this study, especially since another part of it is focusing on counternarratives and prioritizing story-telling – but I would provide more explanation about how CRT can be used to “interrogate Eurocentric and ethnocentric ways in psychological research.” If the authors have more space in the Intro, I'd recommend providing a few specific examples in this paragraph. • Add in a note about when data collected ended, since the pandemic hit around March 2020 (which certainly influenced students' mental health globally) – how do authors plan to account for pandemic-related mental health concerns among Black students? • Provide more information about the three higher education providers that were chosen. • Meaning is unclear here – “In order to generate a narration that has not been structured by the study team's research questions, but by the student's structure of relevance, students are invited to participate in an adapted version of” – how would the narration not be structured by the research questions? • Authors could also consider doing member checking as another measure of reliability after they've started writing up the analysis. • Why do authors think it is useful to have a comparison group in the future? How do they plan to modify the questions to make them relevant to all of these other student groups? Would generally advise removing that statement and focus more on how
---

	interview protocols may focus more intensively on Black UK students' experiences – what are other angles that the current protocol is not tapping into?
--	---

VERSION 1 – AUTHOR RESPONSE

Reviewer: 1

Dr. Seanna Leath, University of Virginia

Comments to the Author:

- I'd reconsider some of the examples that authors offer as contributing to mental health problems – mainly just the “new social connections” part – the others make sense without further explanation

Thank you for this comment, this example has now been amended (pg. 3), and hope this point reads better now.

- CRT does seem useful for this study, especially since another part of it is focusing on counternarratives and prioritizing story-telling – but I would provide more explanation about how CRT can be used to “interrogate Eurocentric and ethnocentric ways in psychological research.” If the authors have more space in the Intro, I'd recommend providing a few specific examples in this paragraph.

Thank you for this comment, we have amended this sentence (pg.4) to be more specific and hope this point reads better now.

- Add in a note about when data collected ended, since the pandemic hit around March 2020 (which certainly influenced students' mental health globally) – how do authors plan to account for pandemic-related mental health concerns among Black students?

We have included the end date for data collection on pages 2, 5 and 7. This is a very good point, but as this study is about exploring students' life journeys and the interpretations of their experience we do not need to account for the pandemic as students will or will not talk about these experiences based on their personal beliefs or perceptions about the pandemic's effect on their mental health.

- Provide more information about the three higher education providers that were chosen.

Thank you for this suggestion, we have provided a brief description of the higher education providers chosen on page 6, and believe this context helps the readability of the manuscript.

- Meaning is unclear here – “In order to generate a narration that has not been structured by the study team's research questions, but by the student's structure of relevance, students are invited to participate in an adapted version of” – how would the narration not be structured by the research questions?

We agree this statement is confusing and have re-worded in the manuscript on page 7.

- Authors could also consider doing member checking as another measure of reliability after they've started writing up the analysis.

This is an excellent suggestion however due to resources constraints, specifically time, due to this study being a PhD project we are unable to complete member checking for all participants.

- Why do authors think it is useful to have a comparison group in the future? How do they plan to modify the questions to make them relevant to all of these other student groups? Would generally advise removing that statement and focus more on how interview protocols may focus more intensively on Black UK students' experiences – what are other angles that the current protocol is not tapping into?

This is an important reflection. We have taken your advice and removed this recommendation and instead expanded on how we can better understand the Black UK students' experiences on page 11-12.